# m6A Methylation-Mediated Stabilization of LINC01106 Suppresses Bladder Cancer Progression by Regulating the miR-3148/DAB1 Axis

**DOI:** 10.3390/biomedicines12010114

**Published:** 2024-01-05

**Authors:** Jun Liu, Cong Tian, Jiajia Qiao, Keming Deng, Xiongjun Ye, Liulin Xiong

**Affiliations:** 1Department of Urology, Peking University People’s Hospital, Beijing 100034, China; hmuliujun@163.com (J.L.); caocongcong0524@163.com (C.T.);; 2The Second Clinical Medical College, Lanzhou University, Lanzhou 730000, China; dengkm19@lzu.edu.cn; 3Department of Urology, National Cancer Center/National Clinical Research Center for Cancer/Cancer Hospital, Chinese Academy of Medical Sciences and Peking Union Medical College, Beijing 100021, China

**Keywords:** bladder cancer, LINC01106, CRISPR/Cas13b, m6A methylation, DAB1

## Abstract

Background: The pivotal roles of long noncoding RNAs (lncRNAs) in the realm of cancer biology, inclusive of bladder cancer (BCa), have been substantiated through various studies. Remarkably, RNA methylation, especially m6A modification, has demonstrated its influence on both coding and noncoding RNAs. Nonetheless, the explicit impact of RNA methylation on lncRNAs and its subsequent contribution to the progression of BCa remains to be elucidated. Methods: In the present investigation, we scrutinized the expression and m6A methylation status of LINC01106, employing quantitative real-time PCR (qRT–PCR) and methylated RNA immunoprecipitation (MeRIP)-qPCR. To decipher the regulatory mechanism underpinning LINC01106, we utilized RNA immunoprecipitation (RIP)-qPCR, methylated RNA immunoprecipitation (MeRIP) assays, and bioinformatic analysis. Furthermore, the CRISPR/dCas13b-METTL3-METTL14 system was implemented to probe the function of LINC01106. Results: The findings of our study indicated that LINC01106 is under expressed and exhibits diminished m6A methylation levels in BCa tissues when compared those of normal controls. A diminished expression of LINC01106 was associated with a less favorable prognosis in BCa patients. Intriguingly, CRISPR-mediated hypermethylation of LINC01106, facilitated by dCas13b-M3-M14, abolished the malignant phenotype of the BCa cells, an effect that could be inverted by Disabled-1 (DAB1) knockdown. From a mechanistic standpoint, we identified an m6A modification site on LINC01106 and highlighted YTHDC1 as a potential reader protein implicated in this process. Additionally, a positive correlation between DAB1 and LINC01106 expression was observed, with miR-3148 potentially acting as a mediator in this relationship. Conclusions: In summary, our research unveils a suppressive regulatory role of the LINC01106/miR-3148/DAB1 axis in the progression of BCa and underscores the YTHDC1-mediated m6A modification mechanism in regards to LINC01106. These revelations propose a new therapeutic target for the management of BCa.

## 1. Introduction

Bladder cancer (BCa) comprehensively impacts the urogenital system, demonstrating an alarming increase in recurrence and mortality rates [1,2,3,4]. Various elements, such as smoking, chemical exposure, and aberrant gene activity, are pivotal in BCa onset and development [5]. Although strides have been made in BCa management, survival outcomes for those at high risk are still not optimal [6,7]. The pivotal influence of gene dysregulation on BCa advancement has been underscored by recent findings [8,9], thereby highlighting the use of targeted therapeutic strategies as a hopeful path for managing advanced BCa cases [10]. Therefore, pinpointing more potent therapeutic targets is vital to augment BCa management outcomes.

The evolution of high-throughput sequencing and analytical technologies has shown that a mere 2% of all transcribed sequences are protein-encoding, revealing that the bulk of transcripts are non-coding RNAs (ncRNAs) [11]. Long non-coding RNAs (lncRNAs), a subclass of ncRNAs, exceed 200 nucleotides in length and have been identified as significant actors in numerous crucial biological activities, such as cell proliferation, apoptosis, migration, and invasion [12,13,14]. Particularly in lung cancer, accumulating evidence indicates that misregulated lncRNAs play a role in modulating various biological functions and may serve as diagnostic markers or therapeutic targets [15]. For example, the lncRNA HOX transcript antisense RNA has been shown to suppress gene expression by attracting chromatin modifiers, thereby enhancing lung cancer cell proliferation, migration, invasion, and drug resistance [16]. In epithelial ovarian cancer, LINC01116 has been associated with advancing the disease by altering the cellular apoptosis pathways [17]. Additionally, LINC01106 has been identified as a promoter of growth and stemness in colorectal cancer by forming a positive feedback loop that controls the Gli family factors [18]. Given LINC01116’s extensive oncogenic roles in various cancers, its potential role in BCa piques our interest, leading us to further explore LINC01116’s involvement in BCa and deepen our understanding of its functional relevance in this setting.

MicroRNAs (miR/miRNA) are a class of noncoding RNAs that function as key regulators in the posttranscriptional modulation of gene expression, both under typical physiological conditions and in various disease states [19]. Increasing evidence suggests a critical involvement of miRNAs in the process of tumorigenesis [20,21]. However, the functions and mechanisms of the majority of miRNAs in cancer biology remain largely unexplored. The specific role of miR-3148 in cancer development is not well-defined. Preliminary research indicates that miR-3148 may act as an oncogenic factor in the colon cancer cell line HCT116, promoting cellular proliferation and aiding in cellular resistance to stress via the MAPK/ERK signaling pathway [22].

Furthermore, Disabled 1 (DAB1), which mirrors the disabled gene in Drosophila, has been recognized for its involvement in neuronal migration and cortical layer formation during cerebral cortex development. Notably, a reduction in DAB1 expression has been documented in primary tumors and various cancer cell lines, with a particular emphasis on brain and endometrial cancers [23]. The work of Huang et al. underscores the pivotal role of DAB1 in the Reelin signaling pathway, highlighting its function as a critical adaptor protein that orchestrates neurodevelopmental processes [24]. In the context of breast cancer, a notable decrease in DAB1 expression has been linked to the triple-negative breast cancer subtype, along with associations to poor differentiation and lymph node metastasis [25].

N6-methyladenosine (m6A) RNA methylation, a common internal modification seen in eukaryotic messenger RNAs (mRNAs), represents about 50% of all methylated ribonucleotides and 0.1–0.4% of total adenosines in cellular RNAs [26]. Recent studies using m6A-specific immunoprecipitation (MeRIP-Seq) have identified over 300 noncoding RNAs that undergo m6A modification in humans and mice. Moreover, it has been recognized that m6A modification in noncoding RNAs plays a vital role in numerous essential biological processes, including stem cell maintenance, cellular differentiation, tissue development, response to heat shock or DNA damage, maternal-to-zygotic transition, primary microRNA processing, and RNA–protein interactions [27]. Given that lncRNAs are transcribed and modified in a manner similar to mRNAs (though typically displaying a higher frequency of m6A sites), it is conceivable that m6A RNA methylation may actively regulate lncRNAs in the same way that it regulates mRNAs, potentially involving analogous mechanisms.

Characterized by its dynamic and reversible characteristics, the N6-methyladenosine (m6A) RNA modification primarily takes place within the consensus motif “RRm6ACH” (R = G or A; H = A, C, or U) and is managed by m6A WERs (“writers”, “erasers”, and “readers”), with the methyltransferase METTL3 being a notable m6A “writer” [28,29]. A burgeoning body of evidence has validated the role of m6A in various cancers. Nevertheless, the explicit role of m6A in bladder cancer (BCa) remains unclear, necessitating further exploration to clarify the dysregulation of METTL3-mediated m6A modification in BCa progression.

This study sought to explore the expression pattern, biological roles, and m6A-mediated regulatory mechanism of LINC01106 through in vitro experiments and bioinformatic analysis. Specifically, we aimed to scrutinize the role of the m6A reader protein YTHDC1 in regulating LINC01106.

## 2. Methods

### 2.1. Patients

The research protocol was conducted in compliance with the principles of the Declaration of Helsinki. Prior to the commencement of the study, written consent was secured from all participating patients. A total of 30 paired tissue samples, both malignant and adjacent normal samples, were procured from BCa patients undergoing surgical procedures.

### 2.2. Cell Culture and Plasmid Transfection

For this research, we utilized a selection of BCa cell lines, namely T24, BIU-87, UMUC3, and 5637, with SVHUC-1 cells serving as the standard reference. These cell lines were sourced from the American Type Culture Collection (ATCC, Manassas, VA, USA) and China’s National Infrastructure of Cell Line Resource. Standard culture conditions for the BCa cell lines involved RPMI 1640 or DMEM enriched with 10% fetal bovine serum (Invitrogen, Carlsbad, CA, USA), and the samples were incubated at 37 °C with 5% CO_2_. For introducing plasmids and siRNAs into the cells, lipo3000 (Invitrogen) was employed, as per the provided guidelines, with a consistent plasmid quantity of 1 μg for experimental purposes.

### 2.3. RNA-Binding Protein Immunoprecipitation (RIP)

The Magna RIPTM RNA-Binding Protein Immunoprecipitation Kit (Millipore) was employed for RIP assays, adhering to the provided protocol. In brief, post cell collection, they were lysed using a comprehensive RIPA buffer, enriched with protease and RNase inhibitors. YTHDC1 antibodies (Abcam, Cambridge, UK) or anti-Argonaute 2 (Ago2) antibodies (Millipore, Billerica, MA, USA) (5 μg) underwent a pre-incubation phase with protein A/G magnetic beads in an immunoprecipitation buffer for two hours, followed by an overnight incubation with cell lysate at 4 °C under gentle rotation. RNA was then separated from the beads, submitted to ethanol precipitation, and subsequently dissolved in RNase-free water. The enrichment of specific RNA segments was determined via qRT-PCR.

### 2.4. Total RNA Isolation and Quantitative Real-Time Reverse Transcription PCR (qRT-PCR)

To isolate RNA from the bladder cancer (BCa) tissues and cell lines, we employed the RNA-easy Isolation Reagent (Vazyme Biotech, Nanjing, China), following the established protocols. For the synthesis of complementary DNA, we utilized 1 μg of total RNA with the HiScript II Q Select RT SuperMix (Vazyme Biotech, Nanjing, China). To assess the expression levels of mature microRNAs, we implemented stem-loop quantitative reverse transcription PCR (qRT-PCR). This process commenced with the binding of stem-loop RT primers to the miRNA molecules, followed by the synthesis of cDNA using reverse transcriptase. Subsequent amplification involved an miR-328-3p-specific forward primer and a universal reverse primer. Quantitative PCR was conducted using the ABI Prism 7500TM system (Applied Biosystems, Foster City, CA, USA) and the universal SYBR Green qPCR Master Mix (Vazyme Biotech, Nanjing, China). Throughout this process, GAPDH was utilized as the normalization control.

### 2.5. M6A RNA Immunoprecipitation (MeRIP) Assay

MeRIP was conducted utilizing the Magna MeRIP m6A Kit (Qiagen, Dusseldorf, Germany), in strict accordance with the guidelines provided by the manufacturer. Initially, protein A/G magnetic beads were conjugated with 3 μg of anti-m6A antibody (Synaptic Systems, Goettingen, Germany), and this mixture was incubated overnight at a temperature of 4 °C. Subsequently, the beads, now conjugated with the antibody, were immersed in IP buffer that included both RNase and protease inhibitors. This step facilitated the binding of the antibody to the target RNAs. Next, the RNAs bound to the antibody were extracted, and their presence and abundance were quantified through qRT-PCR.

### 2.6. RNA Pull-Down Assay

For the RNA pull-down assay, cell nuclear extracts were first incubated with biotin-labeled RNA probes, followed by the addition of streptavidin-coated magnetic beads. This mixture was then allowed to incubate, facilitating the binding of RNA-binding proteins to the biotinylated RNA. Post-incubation, these beads were separated using centrifugal force. After a series of washes to remove non-specifically bound material, the RNA-protein complexes were eluted from the beads. The final step typically involves analyzing these complexes, often through SDS-PAGE and subsequent Western blotting, to identify and characterize the proteins that are interacting with the RNA. The anti-DAB1 antibody was purchased from Millipore (AB5840, 1:1000).

### 2.7. Expression Plasmids, Short Interfering RNAs, and Lentivirus Transfection

The CRISPR dCas13b vectors and Cas13b-gRNA vectors were sourced from Addgene. Synbio Technologies Company (Suzhou, China) was responsible for constructing the miR-3148 mimic, gRNAs, and the dCas13b-M3M14 vector. For overexpression vectors, the coding sequences (CDS) of METTL3, DAB1, and YTHDC1 were integrated into the pcDNA3.1 vector. The pcDNA3.1 vector without any gene insertions was used as a control for subsequent evaluations.

### 2.8. Luciferase Reporter Assay

The BCa cells were transfected with pmirGLO-LINC01106-WT, pmirGLO-LINC01106-Mut, pmirGLO-DAB1-WT, and pmirGLO-DAB1-Mut constructs for one day. The Dual-Glo Luciferase Assay system (Promega Corporation, Madison, WI, USA) was used to measure both firefly (F-luc) and Renilla (R-luc) luciferase activities. The R-luc activity was the normalization factor for the F-luc activity, which was used to gauge reporter transcription. This procedure was performed in triplicate, yielding consistent outcomes.

### 2.9. mRNA Stability

To gauge RNA stability in cells with different vector transfections, the cells were exposed to actinomycin D (Act-D, Catalog #A9415, Sigma-Aldrich, St. Louis, MO, USA) at 5 μg/mL to inhibit transcription. At designated intervals, cells were collected, and RNA was isolated for subsequent real-time PCR analysis. The half-life (t1/2) of the LINC01106 transcript was determined using the formula ln2/slope, with GAPDH serving as the reference gene for normalization.

### 2.10. Cell Proliferation and Apoptosis Assays

The CCK-8 assay (Transgen, Beijing, China) was employed to measure cell growth. The cells (3 × 10^3^) were placed in 96-well plates. At various time points post incubation, the cells were treated with CCK-8 reagent and incubated at 37 °C for 3 h. Absorbance at 450 nm was then recorded using a microplate reader. For apoptosis analysis, the cells with vector transfections were placed in 12-well plates (2 × 10^5^ cells/well). After two days, the caspase-3 levels were detected using a caspase-3/ELISA kit (Hcusabio, Wuhan, China). This procedure was independently repeated in triplicate to ensure consistency.

### 2.11. Wound Healing Migration Assays

Standard protocols were followed for the wound healing migration assays. The cells were seeded at 5 × 10^5^ cells/well in a six-well chamber slide. A sterile pipette tip created a central scratch. At 48 h post incubation, images were captured to assess wound closure. The wound closure percentage was calculated using a specific formula.

### 2.12. Fluorescence In Situ Hybridization (FISH)

T24 and 5637 cells were placed on glass coverslips, washed with PBS (Sigma-Aldrich, St. Louis, MO, USA), fixed with 4% formaldehyde (Sigma-Aldrich) for 30 min, and permeabilized with 70% ethanol (Sigma-Aldrich) overnight. The cells were then exposed to the LINC01106 probe (GenePharma, Shanghai, China) at 37 °C overnight. Post staining with DAPI (Sigma-Aldrich) for 10 min, the cells were washed and visualized using a fluorescence microscope (Olympus, Tokyo, Japan).

### 2.13. Subcellular Fractionation

The Norgen kit (Thorold, ON, Canada) was used to separate cytoplasmic and nuclear RNA fractions. The expression levels of GAPDH, U6, or LINC01106 in the nuclear and cytoplasmic fractions of the T24 or 5637 cells were quantified using qRT-PCR.

### 2.14. Bioinformatic Analysis

To investigate the potential biological functions of LINC01106, we utilized the StarBase online platform (available at https://starbase.sysu.edu.cn/index.php (accessed on 17 January 2023)) for predicting its competing endogenous RNAs (ceRNAs). This approach enabled us to conduct a thorough enrichment analysis of LINC01106. Additionally, for identifying genes in bladder cancer tumors that exhibit expression patterns similar to those of LINC01106, we employed the GEPIA 2 tool (accessible at http://gepia2.cancer-pku.cn/#analysis (accessed on 17 May 2023)). Through this analysis, DAB1 and YTHDC1 emerged as two protein-coding genes showing a positive correlation with LINC01106 expression.

### 2.15. Statistical Analysis

Data are presented as the mean ± standard deviation (SD). Differences between groups were determined using the Student’s *t*-test or ANOVA. A *p*-value below 0.05 was deemed as significant. All tests were conducted in triplicate, using Prism 5 software (GraphPad, La Jolla, CA, USA). Significance was set at * *p* < 0.05, ** *p* < 0.01; NS indicates no significance.

## 3. Results

The downregulation of LINC01106 is associated with poor prognosis and decreased m6A methylation modification levels in RCC.

Initially, we observed a notable decrease in LINC01106 expression in 30 paired BCa samples compared to their adjacent normal counterparts (Figure 1A). Further analysis using the GEPIA online database revealed a strong correlation between reduced LINC01106 levels and unfavorable BCa patient outcomes (Figure 1B, Appendix A). Based on MeRIP-seq data from other research groups [30], BCa tumor samples displayed diminished m6A peak enrichment in LINC01106 transcripts compared to that of adjacent normal samples (Figure 1C). This observation was further validated by MeRIP-qPCR, which confirmed a significant reduction in LINC01106’s m6A levels in tumor samples (Figure 1D). Moreover, qRT-PCR results corroborated the decreased LINC01106 levels in BCa cell lines (T24, BIU-87, and 5637) (Figure 1E). Additionally, MeRIP-qPCR results further confirmed the reduced m6A methylation levels of LINC01106 in T24 and 5637 BCa cells compared to those in normal cells (Figure 1F). Using subcellular fractionation and FISH assays, we determined that LINC01106 predominantly resides in the cytoplasm (Figure 1G,H). These findings prompted a deeper exploration into the relationship between LINC01106 expression, its m6A methylation status, and BCa progression.

### 3.1. m6A Regulates LINC01106 Stability in BCa Cells

To uncover the mechanisms by which m6A modification influences LINC01106 expression, we identified an m6A methyltransferase associated with LINC01106. TCGA pan-cancer correlation analysis revealed a significant positive association between METTL3 and LINC01106 in BCa samples (Figure 2A, Appendix A). Subsequent RT-qPCR results indicated elevated LINC01106 levels in METTL3-overexpressing T24 and 5637 cells (Figure 2B). MeRIP-qPCR further confirmed that METTL3 overexpression significantly augmented m6A-modified LINC01106 transcripts in these cells (Figure 2C,D). Additionally, upon treating METTL3-overexpressing BCa cells with Act-D to halt transcription, RNA stability assays revealed that METTL3 overexpression extended the half-life of the LINC01106 transcript (Figure 2E,F), suggesting that m6A modification might decelerate LINC01106 degradation in BCa cells.

### 3.2. LINC01106 Directly Binds with YTHDC1 in BCa

To further elucidate how m6A modification affects LINC01106 stability, we explored the potential involvement of m6A reader proteins. Prior studies have shown that m6A modification can modulate transcript stability via interactions with specific reader proteins, including IGF2BP1-3 and YTHDC1-2. Notably, a strong positive correlation was observed between LINC01106 and YTHDC1 expression in BCa samples from the TCGA database (Figure 3A). To validate the m6A-dependent role of YTHDC1 in enhancing LINC01106 RNA stability, we employed RIP-qPCR assays. The results highlighted a significant interaction between YTHDC1 and LINC01106 in the BCa cells, which was further enhanced in the METTL3-overexpressing cells (Figure 3B). Overexpressing YTHDC1 in BCa cells led to an increase in LINC01106 levels (Figure 3C). Moreover, YTHDC1 was found to significantly enhance LINC01106 stability in the BCa cells (Figure 3D,E). These findings underscore the role of YTHDC1 in modulating LINC01106 stability through m6A methylation.

### 3.3. Targeting m6A Methylation of LINC01106 by CRISPR/dCas13b-M3-M14 to Regulate BCa Cells Proliferation and Apoptosis

In our study, we utilized a fusion protein, dCas13b-M3-M14, which combines the catalytically inactive Cas13b enzyme with the m6A methyltransferases METTL3 and METTL14 to specifically methylate LINC01106’s m6A site [31]. Targeted guide RNAs were designed to direct this fusion protein to specific regions around LINC01106’s m6A site (Figure 4A). MeRIP-qPCR analysis confirmed the successful methylation of LINC01106 by dCas13b-M3-M14 in T24 and 5637 BCa cells (Figure 4B). This targeted methylation led to a significant upregulation of LINC01106 expression in these cells (Figure 4C). This increased expression was likely due to enhanced binding between LINC01106 and the m6A reader protein YTHDC1, as indicated by our results (Figure 4D). Further functional assays revealed that targeting LINC01106 with dCas13b-M3-M14 significantly impacted BCa cell proliferation, migration, and apoptosis (Figure 4E–I).

### 3.4. DAB1 Is Modulated by LINC01116 in BCa

After elucidating the functional role of LINC01106 in BCa, we proceeded to investigate its regulatory mechanism. The competing endogenous RNA (ceRNA) process has been widely recognized as a post-transcriptional regulatory mechanism. Expanding on this notion, we aimed to identify downstream targets of LINC01106. Utilizing GEPIA 2 analysis, we identified genes that exhibited similar expression patterns to those of LINC01106 in BCa tumors, and DAB1 emerged as the top protein-coding gene positively correlated with LINC01106 (Figure 5A and Appendix A). Previous evidence has highlighted the significant involvement of DAB1 in cancer development [25,32]. To further validate the relationship between LINC01106 and DAB1, we performed Ago2-RIP assays. The results unequivocally demonstrated the enrichment of both LINC01106 and DAB1 in Ago2-associated complexes in T24 and 5637 cells (Figure 5B,C), indicating their coexistence within the RNA-induced silencing complexes (RISCs). Subsequently, we employed RT-qPCR to examine DAB1 expression in BCa cells. The results revealed a distinct increase in DAB1 expression in both T24 and 5637 cells upon LINC01106 activation (Figure 5D). Based on these findings, we postulated that DAB1 functions downstream of LINC01106 in BCa. To further investigate the functional relationship between DAB1 and LINC01106, we conducted luciferase reporter gene assays. The results demonstrated no significant alteration in the relative luciferase activity of the DAB1 promoter in T24 and 5637 cells transfected with small guide RNA (sgRNA) targeting LINC01106. However, the relative luciferase activity of the DAB1 3′ untranslated region (UTR) significantly increased in T24 and 5637 cells with LINC01106 activation (Figure 5E,F). Collectively, these findings suggested a positive correlation between DAB1 and LINC01106 in BCa cells.

### 3.5. LINC01106 Competes with DAB1 in Interacting with miR-3148 in the RISCs

To elucidate LINC01106’s regulatory impact on DAB1, we identified potential miRNAs that could target LINC01106. StarBase analysis identified miR-3148 as a potential miRNA interacting with both LINC01106 and DAB1. Specific binding sites for this interaction were identified (Figure 6A). Luciferase reporter assays confirmed that miR-3148 mimics significantly reduced the luciferase activity of LINC01106-WT/DAB1-WT, but not their mutant counterparts in BCa cells (Figure 6B–E). RNA pull-down assays further supported this interaction (Figure 6F,G). These results suggest that LINC01106 acts as a tumor suppressor, inhibiting BCa cell malignancy, partly through the miR-3148/DAB1 pathway.

### 3.6. Inhibition of DAB1 Reverses the LINC01106 Methylation-Induced Progression Retard in BCa

To further understand LINC01106’s role in regulating DAB1 and its subsequent impact on BCa cell growth, we conducted functional assays. Ago2 RIP assays confirmed the co-localization of LINC01106, miR-3148, and DAB1 within the RISCs in T24 and 5637 cells (Figure 7A,B). Further, we found that LINC01106 methylation increased DAB1 levels, while miR-3148 activation mitigated this increase in T24 and 5637 cells (Figure 7C,D). Functional assays revealed that LINC01106 methylation suppressed BCa cell proliferation, an effect partially reversed by DAB1 knockdown (Figure 7E,F). Moreover, caspase-3 activity analysis and wound healing assays indicated that LINC01106 methylation promoted cell apoptosis and inhibited migration, effects partially rescued by DAB1 downregulation (Figure 7G,H). These findings suggest that LINC01106 acts as a tumor suppressor, inhibiting BCa cell malignancy through m6A methylation modification, partly through the miR-3148/DAB1 pathway.

## 4. Discussion

The pivotal role of long non-coding RNAs (lncRNAs) in diverse cancers, including bladder cancer (BCa), is increasingly being acknowledged [33,34]. Dysregulated lncRNAs are emerging as central actors in BCa’s onset and progression, acting either as oncogenes or tumor suppressors [35,36]. Delving into the function of lncRNAs in BCa offers potential avenues for unveiling novel diagnostic markers and therapeutic strategies. While numerous studies have underscored the significance of LINC01106 in different cancers, its multifaceted roles are evident. For example, LINC01106 has been linked to the amplification of colorectal cancer progression by orchestrating a regulatory feedback loop with Gli family members. In the context of bladder cancer, LINC01106 has been found to influence ELK3 and HOXD8 at the post-transcriptional level, thereby accelerating the disease’s advancement. Moreover, the targeted inhibition of LINC01106 has shown potential in curbing the aggressive traits of gastric cancer cells by interacting with the miR-34a-5p/MYCN pathway. In contrast to these observations, our recent research indicates a notable downregulation and hypomethylation of LINC01106 in BCa cells. Furthermore, the strategic hypermethylation of LINC01106 using CRISPR/dCas13b-M3-M14 markedly curtailed BCa cell proliferation and migration, while enhancing apoptosis. In essence, our findings accentuate LINC01106’s potential as a suppressor in BCa progression, enriching the existing literature on the multifarious roles of lncRNAs in BCa. This research paves the way for the discovery of innovative therapeutic interventions and diagnostic tools for BCa.

Prior research underscores the pivotal role of m6A methylation in modulating various cellular attributes of tumor entities through the regulation of long non-coding RNAs (lncRNAs). For example, in hepatic cancer, the stabilization and subsequent upregulation of LINC00958, facilitated by METTL3-mediated m6A modification, has been implicated in advancing cancer progression [37]. Conversely, the demethylase ALKBH5 is documented to control the expression of KCNK15-AS1 through demethylation, thereby mitigating the migratory and invasive capabilities of pancreatic cancer cells [38]. In the realm of ovarian cancer, YTH domain-containing family protein 1 (YTHDF1) is known to enhance the translation of eukaryotic translation initiation factor 3 subunit C, thereby driving tumorigenesis and metastasis [39]. However, in melanoma, YTHDF1 might function as a tumor suppressor by facilitating the translation of the tumor suppressor gene histidine triad nucleotide-binding protein 2, thereby inhibiting tumor evolution [40]. A previous study has elucidated the vital role of YTHDC1 in recognizing MALAT1-m6A, crucial for sustaining the composition and genomic binding sites of nuclear speckles that regulate the expression of pivotal oncogenes. The artificial anchoring of YTHDC1 to m6A-deficient MALAT1 notably rescues the metastatic potential of cancer cells [41]. In the current study, we unveiled the presence of m6A modification of LINC01106 and its potential binding to YTHDC1, which enhances its stability. By intervening with m6A modification, utilizing the CRISPR/dCas13b-M3-M14 system, we adeptly regulated the abundance of LINC01106 in BCa cells. This evidence suggests that LINC01106 may be regulated by m6A modification and its interaction with YTHDC1. In conclusion, our study augments the expanding literature, emphasizing the regulatory role of m6A modification and the participation of YTHDC1 in the stability and expression of lncRNAs, illustrated by the regulatory interplay between LINC01106, m6A modification, and YTHDC1 in BCa cells.

In our exploration, we discerned that LINC01106 predominantly resides in the cytoplasm of BCa cells. It has been well-established that cytoplasmic lncRNAs typically wield their influence by modulating mRNA stability, translation, or protein alteration [42]. Employing GEPIA, we anticipated a tight linkage between LINC01106 and DAB1 and ascertained the affirmative regulatory impact of LINC01106 on DAB1 within BCa cells. Additionally, we showcased that the methylation of LINC01106 amplified the activity of DAB1 mRNA’s 3′ untranslated region (UTR), signifying the post-transcriptional modulation of DAB1 by LINC01106. A notable post-transcriptional mechanism utilized by lncRNAs reveals their function as competing endogenous RNAs (ceRNAs) in regulatory networks, entailing interactions among lncRNAs, miRNAs, and target mRNAs [34]. During our probe, we pinpointed miR-3148 as a mutual miRNA between LINC01106 and DAB1. Furthermore, we validated that DAB1 was a direct target of miR-3148, and LINC01106 facilitated DAB1 expression in BCa cells. Prior studies have implicated DAB1 in the progression of various cancers, such as prostate cancer [32], colorectal cancer [43], and breast cancer [25]. In our exploration, we unveiled that the impacts of LINC01106 methylation on BCa cells were partially mitigated by the downregulation of DAB1. Therefore, we have elucidated that miR-3148 and LINC01106 competitively bind to the mRNA of DAB1. This competitive interaction plays a significant role in the regulation of DAB1 expression. LINC01106, when modified by m6A, shows an enhanced affinity for DAB1 mRNA, leading to an upregulation of DAB1 expression. On the other hand, miR-3148 binds to the same sites on the DAB1 mRNA but exerts an inhibitory effect, thereby reducing DAB1 expression. In summary, our findings elucidate that LINC01106 orchestrates the proliferative and migratory capacities of malignant BCa cells via the miR-3148/DAB1 axis.

## 5. Conclusions

In conclusion, this study comprehensively investigated the functional implications and clinical relevance of LINC01106 in BCa cell proliferation and migration through a combination of in vitro experiments and bioinformatics analysis. Moreover, we explored the upstream regulatory mechanism of LINC01106 from an epigenomic perspective and revealed the presence of m6A modification sites within its sequence. Notably, we demonstrated that the “reader” protein YTHDC1 plays a role in the regulation of LINC01106 through m6A mediation. In addition, miR-3148 and LINC01106 compete to bind with DAB1 mRNA. LINC01106 enhances DAB1 expression, whereas miR-3148 reduces DAB1 expression. These findings shed light on the molecular mechanisms underlying BCa and provide valuable insights into the potential development of personalized treatment strategies by targeting the m6A modification of LINC01106.

## Figures and Tables

**Figure 1 biomedicines-12-00114-f001:**
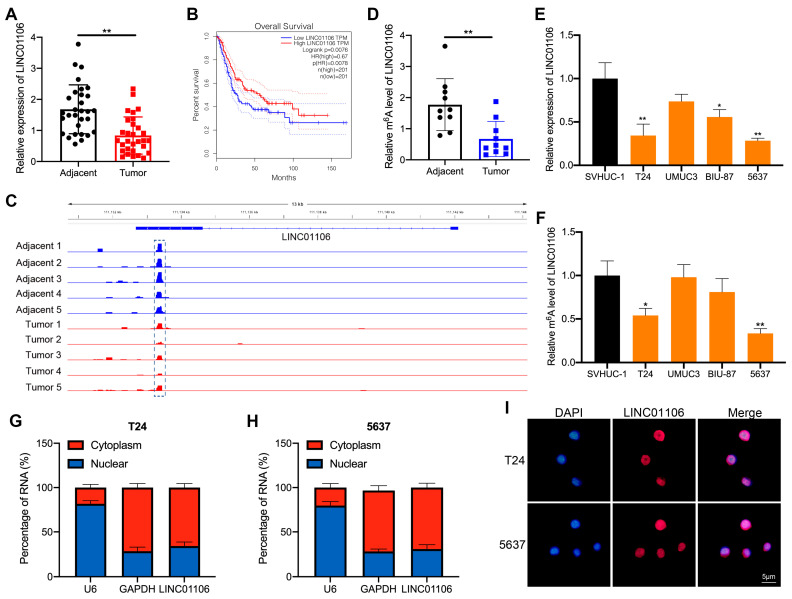
LINC01106 expression is inhibited in BCa tissues and is correlated with the prognosis (**A**). LINC01106 expression was analyzed in 30 BCa tissue pairs using qRT-PCR (**B**). Prognostic outcomes were derived from the average LINC01106 expression in the BCa samples via the GEPIA database to produce a survival graph (**C**). m6A peaks within LINC01106 were visualized with Integrative Genomics Viewer (http://software.broadinstitute.org/software/igv/ (accessed on 17 June 2023)), highlighting m6A sites in five BCa tissue pairs and their adjacent normal counterparts (**D**). MeRIP-qPCR quantified m6A enrichment on LINC01106 in 10 BCa tissue pairs and their adjacent normal tissues (**E**). LINC01106 expression in BCa cell lines (T24, UMUC3, BIU-87, and 5637) was compared to that observed in the SVHUC-1 control cell line using qRT-PCR (**F**). MeRIP-qPCR determined LINC01106’s m6A levels in the BCa cells (**G**–**I**). The cellular location of LINC01106 was ascertained using subcellular distribution and FISH assays, revealing its presence in both the cytoplasm and the nucleus. All results are shown as mean ± SD from three separate tests. Statistical relevance was set at * *p* < 0.05, ** *p* < 0.01.

**Figure 2 biomedicines-12-00114-f002:**
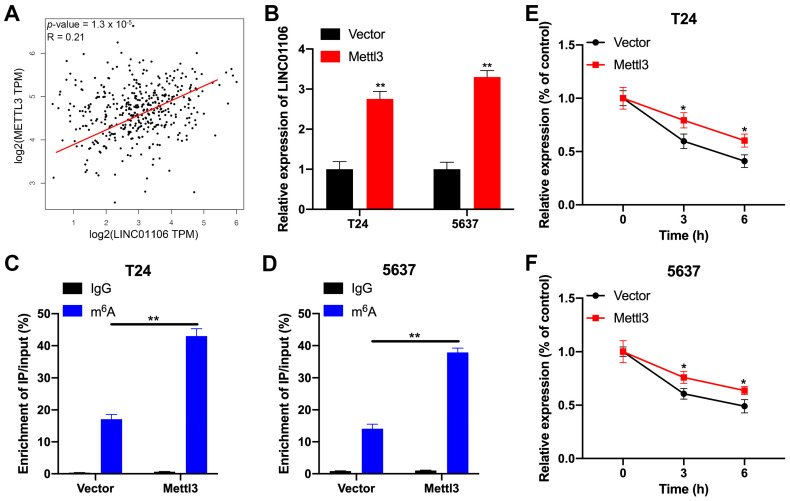
m6A regulates LINC01106 stability in BCa cells (**A**). GEPIA database analysis indicated a positive association between LINC01106 and METTL3 expression in BCa tissues (**B**). At 24 h post transfection of T24 and 5637 cells with either a control vector or Mettl3 construct, LINC01106 expression was quantified using RT-qPCR (**C**,**D**). MeRIP-qPCR identified m6A levels of LINC01106 in T24 (**C**) and 5637 (**D**) cells under standard and Mettl3 overexpression conditions (**E**,**F**). Post Mettl3 overexpression and Actinomycin D (Act-D) treatment, LINC01106 expression in T24 (**E**) and 5637 (**F**) cells was analyzed using RT-qPCR. All results are shown as mean ± SD from three separate tests. Statistical relevance was set at * *p* < 0.05, ** *p* < 0.01.

**Figure 3 biomedicines-12-00114-f003:**
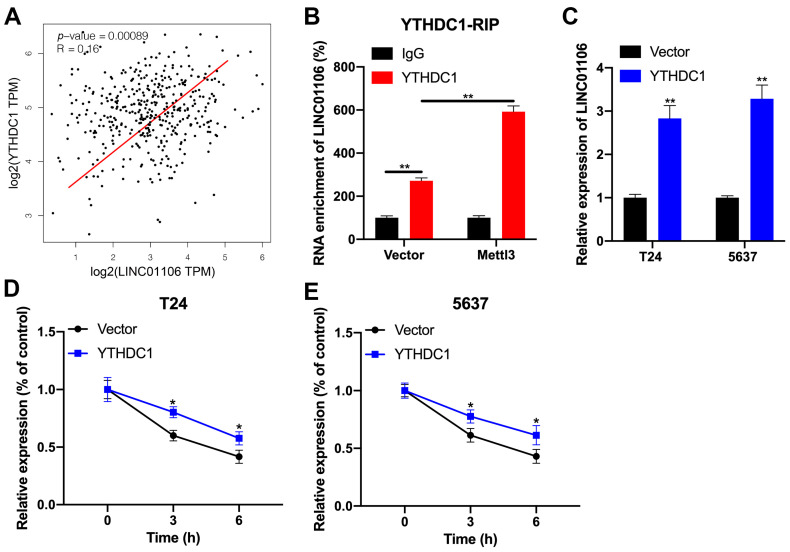
YTHDC1 promotes LINC01106 stability in BCa cells (**A**). GEPIA database analysis of BCa tissues showed a positive association between LINC01106 and YTHDC1 expression (**B**). RIP-qPCR in BCa cells, under standard or Mettl3 overexpression conditions, explored the interaction between LINC01106 and YTHDC1 (**C**). LINC01106 expression in T24 and 5637 cells post-transfection with a control or YTHDC1 vector was determined using RT-qPCR (**D**,**E**). After 24 h transfection with a control or YTHDC1 vector and subsequent Act-D treatment, LINC01106 expression in T24 (**D**) and 5637 (**E**) cells was assessed using RT-qPCR. All results are shown as the mean ± SD from three separate tests. Statistical relevance was set at * *p* < 0.05, ** *p* < 0.01.

**Figure 4 biomedicines-12-00114-f004:**
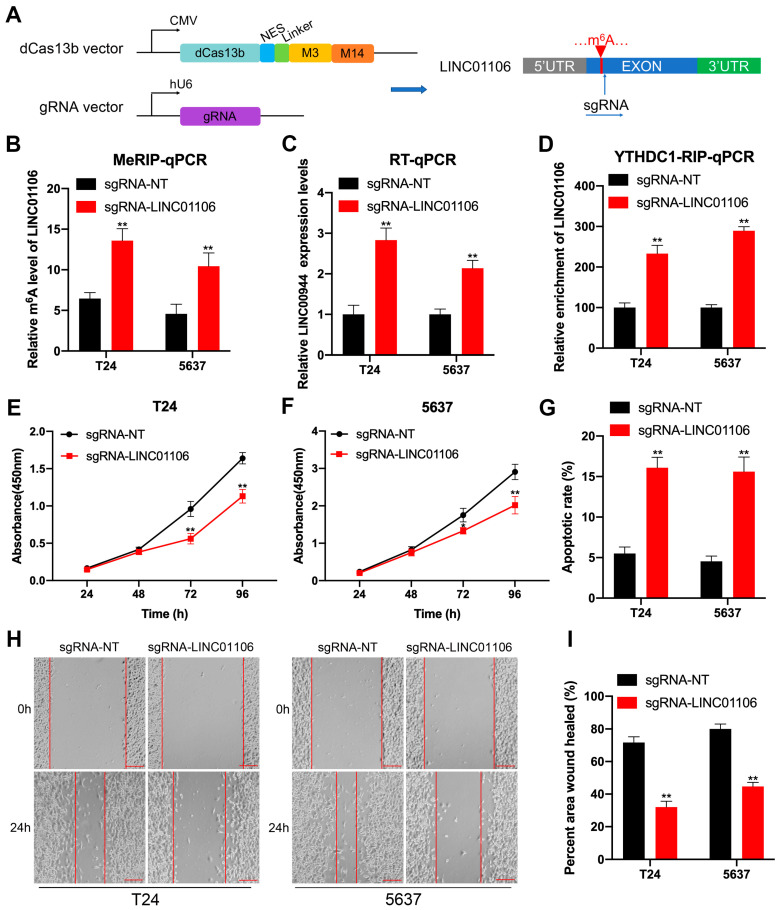
Targeting m6A methylation of LINC01106 by CRISPR/dCas13b-M3-M14 to regulate BCa cell proliferation and apoptosis (**A**). A diagram showcased m6A sites within LINC01106 and the regions targeted by guide RNA (**B**,**C**). After a 24 h transfection with dCas13b-M3-M14 and either a control gRNA or LINC01106-specific gRNA, m6A (**B**) and expression levels (**C**) of LINC01106 in T24 and 5637 cells were evaluated (**D**). Using YTHDC1 antibodies, RIP-qPCR assessed the interaction between LINC01106 mRNA and YTHDC1 in T24 and 5637 cells post 24 h transfection with dCas13b-M3-M14 and either a control gRNA or LINC01106-specific gRNA (**E**,**F**). Cell proliferation in T24 (**E**) and 5637 (**F**) cells post 24 h transfection with dCas13b-M3-M14 and either a control gRNA or LINC01106-specific gRNA was analyzed using a CCK-8 assay (**G**). Caspase-3 ELISA identified cell apoptosis in T24 and 5637 cells post 24 h transfection with dCas13b-M3-M14 and either a control gRNA or LINC01106-specific gRNA. Wound healing assay in T24 and 5637 cells post 24 h transfection with dCas13b-M3-M14 and either a control gRNA or LINC01106-specific gRNA was visualized (**H**) and quantified (**I**). All results are shown as the mean ± SD from three separate tests. Statistical relevance was set at ** *p* < 0.01.

**Figure 5 biomedicines-12-00114-f005:**
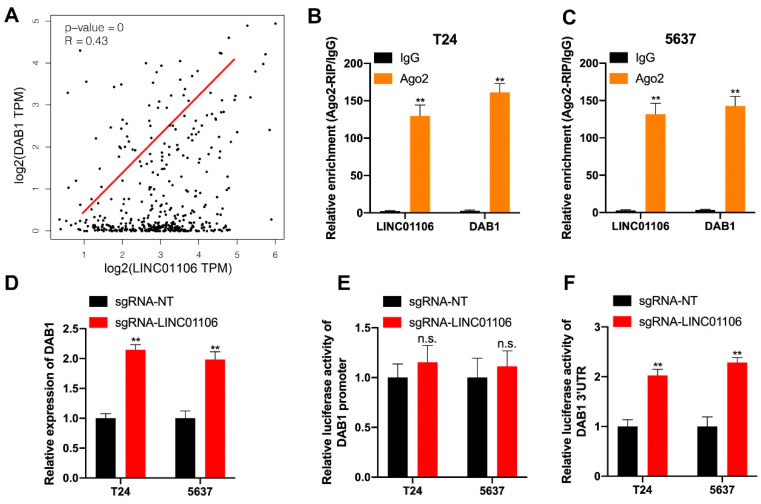
DAB1 and LINC01106 exhibit positive correlations in BCa (**A**). GEPIA database analysis explored the expression relationship between LINC01106 and DAB1 in BCa tissues (**B**,**C**). An Ago2-RIP assay in T24 (**B**) and 5637 (**C**) cells confirmed the presence of LINC01106 and DAB1 in RISCs (**D**). LINC01106 methylation’s impact on DAB1 expression in BCa cells was analyzed using RT-qPCR (**E**,**F**). A dual-luciferase reporter assay further validated the interaction between DAB1 and LINC01106. All results are shown as the mean ± SD from three separate tests. Statistical relevance was set at ** *p* < 0.01, n.s. “not significant”.

**Figure 6 biomedicines-12-00114-f006:**
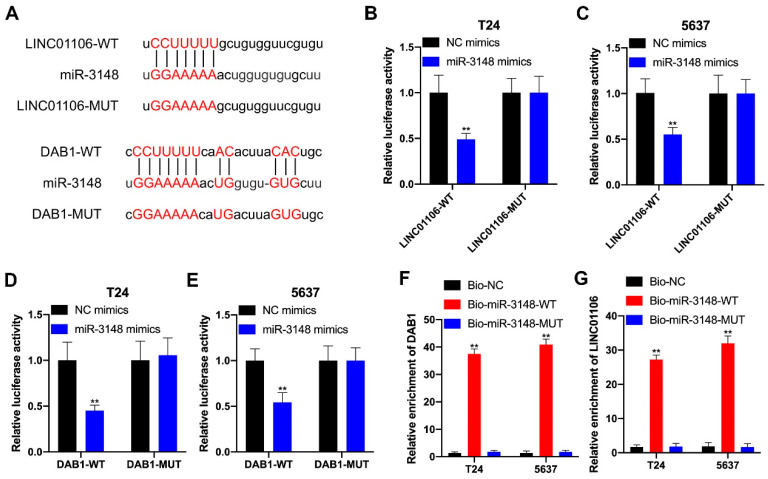
LINC01106, DAB1, and miR-3148 coexist in the same RISC (**A**). StarBase predicted potential miR-3148 binding sites on LINC01106 and DAB1 (**B**,**C**). Luciferase reporter assays in T24 (**B**) and 5637 cells (**C**) predicted miR-3148’s interaction with LINC01106 (**D**,**E**). Luciferase reporter assays in T24 (**D**) and 5637 cells (**E**) predicted DAB1’s interaction with LINC01106 (**F**,**G**). RNA pull-down assays further explored miR-3148’s binding with LINC01106/DAB1. All results are shown as the mean ± SD from three separate tests. Statistical relevance was set at ** *p* < 0.01.

**Figure 7 biomedicines-12-00114-f007:**
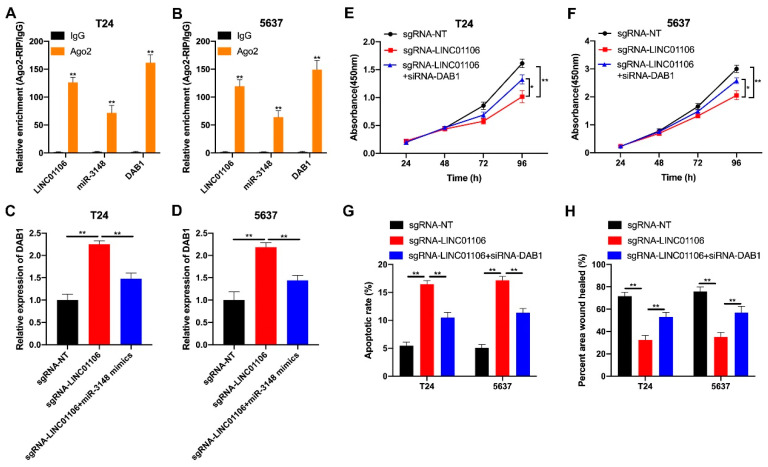
Inhibition of DAB1 reverses the LINC01106 methylation-induced progression retard in BCa (**A**,**B**). The interaction between LINC01106, miR-3148, and DAB1 in RNA-induced silencing complexes (RISCs) was confirmed through Ago2-RIP assays in T24 (**A**) and 5637 cells (**B**) (**C**,**D**). The expression levels of DAB1 were assessed in the T24 (**C**) and 5637 cells (**D**) transfected with sgRNA-NT, sgRNA-LINC01106, or co-transfected with sgRNA-LINC01106 and miR-3148 mimics (**E**,**F**). Cell proliferation was evaluated using CCK8 assays in T24 (**E**) and 5637 cells (**F**) transfected with sgRNA-NT, sgRNA-LINC01106, or co-transfected with sgRNA-LINC01106 and siRNA-DAB1 for 24 h (**G**). Cell apoptosis was detected using a caspase-3 ELISA kit in T24 and 5637 cells transfected with sgRNA-NT or sgRNA-LINC01106, or co-transfected with sgRNA-LINC01106 and siRNA-DAB1, for 24 h (**H**). Quantitative analysis of the wound healing assay was performed in T24 and 5637 cells transfected with sgRNA-NT or sgRNA-LINC01106, or co-transfected with sgRNA-LINC01106 and siRNA-DAB1, for 24 h. All data are presented as the mean ± standard deviation (SD) and were derived from three independent experiments. Statistical significance was determined with a significance level of * *p* < 0.05, ** *p* < 0.01.

## Data Availability

The datasets used and/or analyzed during the current study are available from the corresponding author on reasonable request.

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
